# Advancing Model Refinement: Muon-Optimized Distillation and Quantization for LLM Deployment

## Abstract

Large Language Models (LLMs) are increasingly deployed on resource-constrained edge devices, where 4-bit post-training quantization is a dominant tool for reducing memory footprint. A central but underexplored question is whether the choice of *fine-tuning* optimizer affects how gracefully a model degrades under subsequent aggressive quantization. Recent work has shown that Muon-pretrained models exhibit fewer activation-channel outliers and correspondingly lower accuracy degradation under 4-bit quantization than Adam-pretrained models (Park et al., 2025), but this observation has been confined to the pretraining regime.

In this work, we test whether this quantization robustness extends to the parameter-efficient fine-tuning regime relevant to practical edge deployment. We integrate synthetic data generation, logit-based knowledge distillation from a vocabulary-aligned teacher, LoRA fine-tuning, Bayesian hyperparameter optimization, and GPTQ 4-bit quantization into a single end-to-end pipeline, and use it as a controlled testbed to compare Adam-optimized and Muon-optimized LoRA fine-tuning across eight standard LLM benchmarks, with five HPO replicates per condition for statistical reporting.

We report two primary empirical findings. First, Muon-optimized LoRA fine-tuning yields models that degrade less under 4-bit quantization than Adam-optimized counterparts on six of eight benchmarks, extending the pretraining-era observation of Park et al. (2025) to the fine-tuning regime. Second, Bayesian hyperparameter optimization consistently selects pure KL-divergence alignment with the teacher ($\alpha = 1$), indicating that on synthetic distillation data the teacher's output distribution is the dominant training signal relative to supervised cross-entropy. The full pipeline achieves approximately $2\times$ memory compression (e.g., $6\,\text{GB}$ to $3\,\text{GB}$) and up to 50% lower per-token latency while matching or exceeding naive GPTQ quantization on every benchmark studied.

## 1 Introduction

Recent advancements in Large Language Models (LLMs) have seen their rise across industrial and government applications. Billions of dollars are now spent on LLM training and inference each year across data centers and edge devices, enabling users to complete an expanding array of tasks. However, the size and expense of these models strains the current infrastructure. A perennial question remains: how should researchers and engineers optimize models for efficient inference on specific user tasks?

The inference challenge is compounded on edge devices, where the local hardware imposes strict memory, energy, and network constraints. Authors have implemented a variety of solutions to shrink model memory requirements, reduce compute required during inference, and accelerate inference speed; many of these methods have deep roots in the field (LeCun et al., 1989; Hinton et al., 2015; Gray and Neuhoff, 1998). Model pruning, low-rank approximation, quantization, and distillation all form a family of core model compression methods that decrease memory requirements (Sander et al., 2025). It remains a challenge to effectively combine techniques in each family of methods with each other as the field matures (Zheng et al., 2025; Dong et al., 2025). This fragmentation motivates a unified optimization pipeline, where data generation, training

objectives, optimizer dynamics, and post-training compression are jointly considered to produce models that generalize across downstream tasks under resource constraints.

We also note that in practical deployment settings, optimization is not a one-time process but a continuous adaptation problem. Real-world systems often consist of teams of heterogeneous devices operating under varying hardware constraints, where models must be repeatedly specialized for evolving tasks. For example, in distributed edge environments such as autonomous systems or sensor networks, models may be deployed across hundreds of devices with differing memory, compute, and energy budgets, while simultaneously being re-targeted to new objectives as mission requirements change. Under these conditions, static or task-specific optimization strategies are insufficient; instead, there is a need for streamlined optimization pipelines that can efficiently adapt models across both task distributions and hardware configurations with minimal manual intervention.

We propose and empirically study such an optimization framework for efficient LLM deployment on resource-constrained platforms, and we use it as a controlled testbed to answer a focused scientific question: *does the quantization robustness of Muon-optimized models, previously demonstrated only during pretraining, extend to the LoRA fine-tuning regime?*

This paper makes three contributions:

1. **Extension of Muon quantization robustness to fine-tuning.** We present the first empirical evidence, to our knowledge, that Muon-optimized LoRA fine-tuning produces student models that degrade less under subsequent 4-bit GPTQ quantization than Adam-optimized counterparts, on six of eight standard LLM benchmarks, with statistical reporting across five HPO replicates per condition. This extends the pretraining-era finding of Park et al. (2025) into the parameter-efficient fine-tuning regime most relevant to practical edge deployment.

2. **A controlled pipeline for compression-aware fine-tuning.** We introduce an end-to-end pipeline combining self-instruct synthetic data generation, vocabulary-aligned logit distillation, LoRA fine-tuning, Bayesian hyperparameter optimization, and GPTQ quantization, designed so that each component can be ablated independently. We use this pipeline both as a deployable system and as the experimental harness for the optimizer-compression study above.

3. **Empirical characterization of the distillation loss landscape under synthetic data.** Across all eight benchmarks, Bayesian HPO consistently selects pure KL-divergence alignment with the teacher ($\alpha = 1$), providing evidence that on synthetic distillation data the teacher's output distribution is the dominant training signal relative to supervised cross-entropy. We discuss implications for future synthetic-data distillation pipelines.

The remainder of the paper details the proposed framework, its components, and empirical evaluations demonstrating its effectiveness across multiple benchmarks.

## 2 Background and Related Work

We review key techniques central to model compression pipelines for large language models (LLMs). These include pruning, quantization, low-rank approximation, knowledge distillation, and parameter-efficient fine-tuning via Low-Rank Adaptation (LoRA). We also discuss synthetic data generation, which is used to produce training data that enables knowledge distillation from a stronger model without relying on large human-annotated datasets. Finally, we discuss the recent Muon optimizer, which we integrate into the framework to boost model robustness to quantization error.

**Model compression** schemes aim to reduce the size, inference latency, and computational footprint of LLMs while preserving performance. Common techniques include pruning, low-rank approximations, knowledge distillation, and quantization (Sander et al., 2025). Pruning removes redundant weights or neurons, often guided by magnitude-based criteria or advanced methods like the Lottery Ticket Hypothesis, which identifies sparse subnetworks that match full model accuracy (Frankle and Carbin, 2019). Recent advancements, such

as SparseGPT (Frantar and Alistarh, 2023), enable one-shot pruning for generative models without extensive retraining.

Low-rank approximations decompose high-dimensional weight matrices into lower-rank factors, reducing parameter count without full retraining. Such techniques have been extended to LLMs, where authors utilize Singular Value Decomposition-based schemes (Hsu et al., 2022) (Wang et al., 2024) (Wang et al., 2025).

Quantization reduces the precision of model weights and activations (e.g., from FP16 to INT4), significantly lowering memory usage. GPTQ (Frantar et al., 2023) uses Hessian-based approximations to minimize the reconstruction error introduced by quantization in LLMs. CALDERA (Saha et al., 2024) and QLoRA (Dettmers et al., 2023) combine quantization with low-rank adaptation, achieving efficient compression while preserving model performance through joint optimization. Other methods, like Activation-aware Weight Quantization (AWQ) (Lin et al., 2024), prioritize salient weights to maintain output fidelity.

Knowledge distillation (KD) transfers knowledge from a large "teacher" model to a smaller "student" model to improve performance on downstream tasks (Hinton et al., 2015). Rather than relying solely on supervised, one-hot encoded labels, KD uses the teacher's predictive distribution as a training signal, providing richer supervision that improves generalization. We employ logit-based distillation, where the student model (S1) is trained to match the teacher model (T2) by minimizing the KL divergence between their output distributions. Recent work (Cloud et al., 2025) demonstrates that KL-based distillation can transfer information beyond the explicit training labels, suggesting that the teacher's output distribution encodes additional structure learned during pretraining.

**Synthetic data generation** has become an important technique for reducing reliance on large-scale human-annotated datasets, particularly in the training and fine-tuning of large language models. Under this paradigm, the teacher model is used to generate labeled examples, which are then used to supervise a smaller or more specialized model.

A prominent example is the self-instruct framework (Wang et al., 2023), where LLMs are iteratively prompted to produce diverse instruction-response pairs from a small set of seed prompts. Similar teacher-driven pipelines have been widely adopted to construct instruction-tuning datasets, demonstrating that high-quality synthetic supervision can effectively substitute for curated data.

**Low-Rank Adaptation (LoRA)** is a parameter-efficient fine-tuning (PEFT) technique that adds trainable low-rank matrices into linear layers, while freezing original weights (Hu et al., 2021). Variants like QLoRA (Dettmers et al., 2023) integrate quantization of the model parameters for a lower memory footprint, but include full-precision low-rank matrices as adapters. In distillation pipelines, LoRA complements KD by enabling student models to adapt to teacher-generated data without overfitting or catastrophic forgetting, as supported by frameworks like Hugging Face's PEFT library.

**Hyperparameter optimization (HPO)** plays a critical role in achieving strong performance in modern deep learning systems, particularly for multi-component pipelines where interactions between parameters can be complex. Classical approaches such as grid and random search are simple but inefficient in high-dimensional spaces. More recent work has focused on adaptive methods, including Bayesian optimization (Snoek et al., 2012), which models the objective function to balance exploration and exploitation, and resource-aware methods such as Hyperband (Li et al., 2018). In this work, we employ Bayesian optimization using Optuna (Akiba et al., 2019), which implements a Tree-structured Parzen Estimator (TPE) to efficiently search heterogeneous hyperparameter spaces. Sample efficiency is essential in our setting, where each evaluation involves fine-tuning and compression of large models.

Accuracy preservation after compression is critical for the practical usability of LLMs in resource-constrained deployment settings. Aggressive quantization can introduce degradation in task accuracy. A key source of this degradation is the presence of outlier activations and poorly conditioned weight distributions, which amplify rounding errors during low-bit quantization. Choice of optimizer also affects a model's underlying weight and activation distributions, in turn affecting performance under quantization (Park et al., 2025)

A recent advancement in the optimization of 2d tensors, **Muon** factors the gradients of 2d layers approximately using Newton-Schulz and then performs descent over the spectral norm of each layer (Jordan et al., 2024).

By effectively amplifying the significance of small directions, Muon accelerates learning, achieving 30-40% reduction in training time and tokens required for small models. Further research has shown the solutions that Muon finds are quantitatively different than Adam-optimized solutions, with a lower population of outlier channel activations (Park et al., 2025). Authors find that a smaller domain of channel activations decreases rounding errors during quantization, subsequently increasing compressed model accuracy and decreasing perplexity (Park et al., 2025).

Other authors show a more complicated picture when Adam pre-trained models are used with Muon fine-tuning, and vice versa. They find that depending on which benchmark is tested, Adam pre-training paired with Muon fine tuning gives suboptimal accuracy (Liu et al., 2025).

In addition, we find logit-based distillation losses used in combination with Muon an underexplored topic, with only a single paper on distillation of specific latent features in the vision context (Chen et al., 2025).

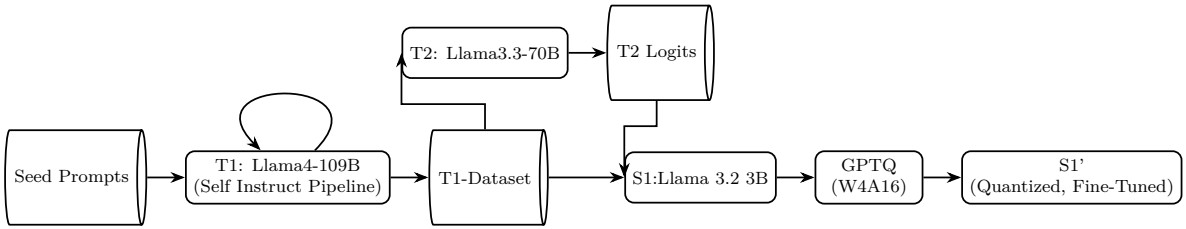

Figure 1: Pipeline Overview

## 3 Methodology

We present an end-to-end framework for compressing a Large Language Model (LLM) and specializing it for a specific task while maintaining performance on resource-constrained edge devices. The framework integrates knowledge distillation, synthetic data generation, low-rank adaptation (LoRA), GPTQ-based compression, and Bayesian hyperparameter optimization, as illustrated in Figure 1. The framework provides a consistent way to fine-tune a compact student model ($S_1$) using knowledge distillation from a knowledge teacher model ($T_2$), on a task-specific dataset generated through synthetic data generation using a high-performance teacher ($T_1$), in conjunction with quantization, and hyperparameter optimization to optimize the $S_1$ for edge deployment.

### 3.1 Framework

To create an efficient, task-specialized model, we first choose a compact student model, $S_1$, and a teacher model, $T_2$, both sharing the same tokenizer (Boizard et al., 2025). The shared tokenizer ensures that the vocabulary spaces of $T_2$ and $S_1$ are aligned, eliminating support mismatch in their output distributions. Support mismatch here will catastrophically inflate the Kullback-Liebler Divergence used as the distillation loss function. For our experiments, we select Meta's Llama 3.1 70B Instruct model as $T_2$ and Llama 3.2 3B Instruct as $S_1$ (AI, 2024a). The compact size of $S_1$ enhances resource efficiency for edge deployment, while $T_2$'s high performance on the target task provides robust, learnable information encoded in the model's output logits.

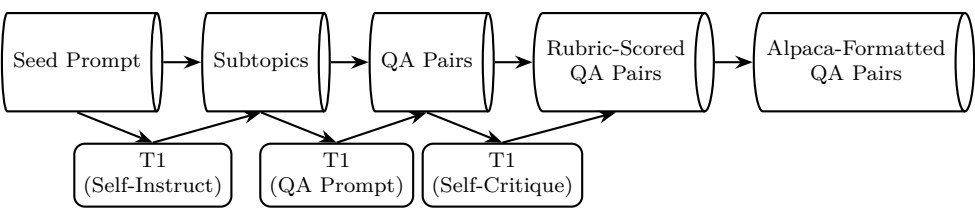

Figure 2: Self-Instruct Pipeline Detail

Our knowledge distillation employs Low-Rank Adaptation (LoRA) for parameter-efficient fine-tuning of $S_1$ (Hu et al., 2021). LoRA parameterizes weight updates using low-rank factors $B \in \mathbb{R}^{d \times r}$ and $A \in \mathbb{R}^{r \times k}$, where $r \ll \min(d, k)$ (typically $r \in [8, 64]$). The pre-trained weights $W \in \mathbb{R}^{d \times k}$ are frozen, and the update is given by $\Delta W = BA$. Optimization is performed only over the low-rank factors $A$ and $B$, reducing memory and computational costs relative to full-rank fine-tuning, while also mitigating overfitting.

The distillation process minimizes a combined loss function balancing Cross Entropy (CE) loss and KLD between $T_2$ and $S_1$ logits. The CE loss is:

$$\mathcal{L}_{\text{CE}}(y_b, p_S) = -\sum_i y_b(i) \log p_S(i), \tag{1}$$

where $y_b$ is the ground-truth label, and $p_S$ is the student's softmax output. The softmax outputs for $T_2$ and $S_1$ are:

$$p_T(i) = \frac{\exp(t_i/T)}{\sum_j \exp(t_j/T)}, \tag{2}$$

$$p_S(i) = \frac{\exp(s_i/T)}{\sum_j \exp(s_j/T)}, \tag{3}$$

where $t_i$ and $s_i$ are the logits of $T_2$ and $S_1$, respectively, and $T$ is the distillation temperature. The KLD loss, which benefits from the shared tokenizer to ensure $p_T$ and $p_S$ operate over the same vocabulary space, is:

$$\mathcal{L}_{\text{KL}}(p_T, p_S) = T^2 \sum_i p_T(i) \log \frac{p_T(i)}{p_S(i)}, \tag{4}$$

and the combined fine tuning loss is:

$$\mathcal{L} = \alpha \mathcal{L}_{\text{KL}}(p_T, p_S) + (1 - \alpha)\mathcal{L}_{\text{CE}}(y_b, p_S), \tag{5}$$

where $\alpha \in [0, 1]$ balances the two terms. We perform a single epoch of fine-tuning with a 16-sample Bayesian Hyperparameter Optimization (HPO) to optimize $\alpha$, LoRA rank, LoRA scaling, distillation temperature, weight decay, and learning rate. As one of our experimental conditions, we use either Adam (Kingma and Ba, 2017) or Muon (Jordan et al., 2024) as the optimizer. After fine-tuning, the LoRA is merged into $S_1$.

While $T_2$ enables task-specific knowledge transfer through vocabulary-aligned distillation, it still requires a task-aligned dataset to supervise training. In many realistic settings, such datasets are limited or unavailable. To address this, we introduce another teacher model, $T_1$, whose role is to synthesize task-specific data rather than directly participate in distillation. We instantiate $T_1$ as a larger, more capable model with broad domain knowledge (Meta's Llama 4 Scout 109B (AI, 2024b)), and use it to generate supervision data via a Self-Instruct-style pipeline (Wang et al., 2023; Álvaro Bartolomé Del Canto et al., 2024). Concretely, $T_1$ expands a set of seed prompts into diverse instruction prompts and corresponding question-answer pairs, producing a synthetic dataset tailored to each task. The pipeline begins with manually designed seed prompts containing task-relevant keywords (e.g., "Astronomy" or "Virology" for MMLU). These seeds are iteratively expanded into subtopics (e.g., "black hole formation" or "viral replication cycles"), from which $T_1$ generates instruction-following examples. We fix inference parameters (temperature 0.7, top-p 0.95) to balance diversity and consistency. To improve data quality, $T_1$ further evaluates generated samples using a rubric-based self-critique, discarding low-quality entries. We target 600 examples per task, with less than 10% removed during filtering. We emphasize that the seed prompts are manually constructed and represent a potential bottleneck in the pipeline. Improving prompt design or automating prompt optimization is a promising direction for future work. The resulting dataset is formatted in Alpaca style (Taori et al., 2023) to preserve the instruction-following behavior of $S_1$ (Figure 2).

The dataset is tailored to specific tasks to ensure $S_1'$'s specialization. We select 8 benchmarks: MMLU, assessing knowledge and reasoning across 57 academic disciplines; ARC-e, testing commonsense reasoning with easy science questions; CommonsenseQA, evaluating textual commonsense reasoning; HellaSwag, benchmarking grounded commonsense inference in physical or social scenarios; OpenBookQA, measuring

Table 1: Accuracies of Pre-Trained Models T1 (Llama4-Scout), T2 (Llama3.3-70B-Instruct), and S1 (Llama 3.2 3B Instruct)

| Task | Llama4Scout | Llama3.3 70B | Llama3.2 3B |
|------|-------------|--------------|-------------|
| ARC-e* | .8384 | .8283 | .7100 |
| CQA | .8239 | .8346 | .7371 |
| HS* | .8265 | .8519 | .7162 |
| MMLU | .7987 | .8221 | .6069 |
| OBQA* | .4580 | .4640 | .3940 |
| PIQA* | .8090 | .8471 | .7682 |
| SIQA | .4744 | .5307 | .4744 |
| WG | .7545 | .8311 | .6875 |

advanced scientific reasoning; PIQA, testing physical commonsense reasoning; SIQA, evaluating social intelligence; and WinoGrande, testing pronoun resolution in challenging contexts. These benchmarks, their respective abbreviations, and their respective citations are given in table 2. For each task, we selected 20 key phrases that span the space of topics in the benchmark. For example, Commonsense Question Answer requires a model to be an expert in "social interactions", "physical causality", "object properties and uses", and other general skills. These key phrases are inserted into a seed prompt template, which are used to guide $T_1$'s Self-Instruct synthetic data generation, in turn producing a diverse dataset for $S_1$'s fine-tuning.

Table 2: Eight language benchmarks considered

| Task | Full Name | Reference |
|------|-----------|-----------|
| ARC-e | ARC (Easy) | (Clark et al., 2018) |
| CQA | CommonsenseQA | (Talmor et al., 2018) |
| HS | HellaSwag | (Zellers et al., 2019) |
| MMLU | MMLU | (Hendrycks et al., 2021) |
| OBQA | OpenBookQA | (Mihaylov et al., 2018) |
| PIQA | Physical Interaction QA | (Bisk et al., 2020) |
| SIQA | SocialIQA | (Sap et al., 2019) |
| WG | WinoGrande | (Sakaguchi et al., 2019) |

To enable edge deployment, we compress $S_1'$ using GPTQ 4-bit quantization (Frantar et al., 2023), producing the final model $S_1''$. GPTQ minimizes quantization error compared to alternatives like round-to-nearest quantization (Xiao et al., 2024), even outperforming AWQ (Lin et al., 2024) in accuracy preservation on some benchmarks. We use w4a16 quantization (4-bit weights, 16-bit activations) on linear layers, achieving approximately 2× memory compression compared to full-precision models, suitable for resource-constrained hardware (Han et al., 2016).

## 3.2 Model Optimization

To address the challenges of deploying LLMs on edge devices, which are limited by memory (often 1-8 GB), compute power, and energy consumption-the optimization process must balance model performance with resource constraints. These devices impose strict limitations on model size, inference latency (ideally sub-100 ms for real-time applications), and power usage, making direct deployment of massive LLMs like Llama-7B (requiring approximately 14 GB in FP16) infeasible without significant adaptations. The optimization problem incorporates non-differentiable hyperparameters, and is also highly non-convex, ruling out traditional optimization solution strategies like gradient descent. One approach involves reformulating the problem into a subproblem or series of subproblems by relaxing constraints. A series of such subproblems decompose the intractable, high-dimensional optimization problem into a sequence of lower-dimensional subproblems.

To optimize the pipeline's performance, we relax the compute power and energy constraints while retaining the memory constraint, described in 3.2.

$$\text{Minimize} \quad \mathcal{L}_{CE}^{val}(\mathbf{a}, \mathbf{r}, \eta, \mathbf{T}, \alpha, \mathbf{w_d})$$
$$\text{subject to:} \quad h(\hat{m}) \leq \text{Memory}_{\text{budget}}$$

where $\mathcal{L}_{CE}^{val}$ denotes the cross-entropy loss evaluated on a held-out validation set using ground-truth labels, as defined in (1).

We optimize hyperparameters using Bayesian optimization implemented via Optuna (Akiba et al., 2019), which employs a Tree-structured Parzen Estimator (TPE) for efficient search. The objective is to minimize validation loss on a 10% held-out split of the T1 dataset. Each trial consists of fine-tuning the student model under a given hyperparameter configuration, followed by evaluation on the validation set. The search space for key hyperparameters is defined in Table 3, including LoRA rank and scaling, learning rate, distillation temperature and weight, and weight decay. We run a fixed number of optimization trials (16) under a constrained compute budget, selecting the configuration that achieves the lowest validation loss. The hyperparameters we consider are described in Table 3 and results summarized in Table 6.

| Variable | Description | Search Space |
|---|---|---|
| $\mathbf{a}$ | LoRA scaling | $[0.5, 2.0]$ |
| $\mathbf{r}$ | LoRA rank | $[8, 64]$ |
| $\boldsymbol{\eta}$ | Learning rate | $[10^{-5}, 10^{-3}]$ |
| $\mathbf{T}$ | Distillation temperature | $[0.5, 8]$ |
| $\boldsymbol{\alpha}$ | Distillation loss weight | $[0, 1]$ |
| $\mathbf{w_d}$ | Weight decay | $[0, 2]$ |

Table 3: Hyperparameter definitions and search spaces used for Bayesian optimization.

We consider two experimental conditions corresponding to the choice of optimizer: Adam (Kingma and Ba, 2017) and Muon (Jordan et al., 2024). For each experimental condition, we run 5 replicates of 16 generations independently to obtain near-optimal hyperparameters. Prior work suggests that Muon-pretrained models exhibit increased robustness to quantization, with smaller post-quantization accuracy degradation (Section 2). We investigate whether this robustness extends to parameter-efficient fine-tuning; after HPO and fine-tuning, we evaluate each model before and after quantization. We define the quantization-induced accuracy drop as the difference between these two accuracy measurements.

We propose a unified framework that combines synthetic data generation via $T_1$, vocabulary-aligned distillation from $T_2$, LoRA-based fine-tuning of $S_1$, and GPTQ compression, with hyperparameters optimized using Bayesian optimization. We further evaluate the use of the Muon optimizer under distillation and LoRA as a mechanism for improving quantization robustness. This produces a compact, task-specialized model $S_1''$ suitable for edge deployment. The framework is designed to be modular and extensible, allowing flexible choices of models, tasks, and quantization schemes. While the current implementation relies on manually designed seed prompts for $T_1$ and a fixed compression scheme, future work will investigate automated prompt optimization, alternative compression methods, and additional regularization terms in the fine-tuning loss to further improve model robustness under compression. The full pipeline is described algorithmically in 1.

---

**Algorithm 1** Pipeline

---

**Require:** Teacher $T$, base model $S$, seed prompts $\mathcal{D}_{\text{seed}}$
**Ensure:** 4-bit quantized student $S''$
1: $\mathcal{D}_{\text{gen}} \leftarrow$ Self-Instruct-with-Rubric($T$, $\mathcal{D}_{\text{seed}}$)          $\triangleright$ seed $\rightarrow$ subtopics $\rightarrow$ questions $\rightarrow$ answers $\rightarrow$ rubric
2: Search space: $\mathbf{a}$, $\mathbf{r}$, $\mathbf{w_d}$, $\eta$, $\mathbf{T}$, $\alpha$
3: $(\hat{a}, \hat{r}, \hat{w}_d, \hat{\eta}, \hat{T}, \hat{\alpha}) \leftarrow$ Optuna-HPO on validation split of $\mathcal{D}_{\text{gen}}$     $\triangleright$ using $\mathcal{L} = \alpha\, D_{\text{KL}}(T \parallel S) + (1 - \alpha)\, \text{CE}$
4: $S' \leftarrow$ LoRA-finetune($S_0$, $\mathcal{D}_{\text{gen}}$, $(\hat{a}, \hat{r}, \hat{w}_d, \hat{\eta}, \hat{T}, \hat{\alpha})$)
5: $\mathcal{D}_{\text{calib}} \leftarrow$ subsample 128 sequences from $\mathcal{D}_{\text{gen}}$
6: $S'' \leftarrow$ GPTQ($S$, 4-bit, group-size=128, calibration=$\mathcal{D}_{\text{calib}}$)
7: **return** $S''$

---

# 4 Results & Discussion

We present our results in distinct sections to interrogate the different framework components. First, we examine the data-generation and fine-tuning process prior to compression. Then, we evaluate accuracies under GPTQ compression and compare how much accuracy is lost under compression. We examine hyperparameters from an HPO replicate to discuss the optimal loss function that HPO discovers. Finally, we examine model size and throughput statistics to confirm model inference acceleration under compression.

## 4.1 Synthetic Data Generation and Distillation Performance

We first verify that the synthetic data generation process is appropriately aligned with each benchmark. To this end, we compare pre-trained Llama3.2-3B accuracies with those of Adam- and Muon-optimized models after fine-tuning but prior to GPTQ compression. We observe only marginal improvements from fine-tuning: both optimizers yield accuracies within one standard deviation across five HPO replicates, indicating limited gains in the uncompressed regime. MMLU exhibits notably higher variance, consistent with its scale and multi-domain structure with 57 subjects. These results suggest that the pre-trained representations are already near performance saturation, and that if we observe any improvements under compression primarily arise from the increased robustness of the fine-tuned minima rather than raw accuracy gains in the fine-tuned model.

Table 4: Comparison of Pre-Trained and Fine-Tuned Accuracies (Mean $\pm$ Std; Max over 5 HPO replicates)

| Task | Llama3.2-3B | Adam-LoRA ($\mu \pm \sigma$; max) | Muon-LoRA ($\mu \pm \sigma$; max) |
|---|---|---|---|
| ARC-e* | .7100 | .6951 $\pm$ .0034; .7125 | .7106 $\pm$ .0034; **.7138** |
| CQA | .7371 | .7369 $\pm$ .0027; .7412 | .7351 $\pm$ .0084; **.7420** |
| HS* | .7162 | .7112 $\pm$ .0079; .7203 | .7152 $\pm$ .0071; **.7220** |
| MMLU | .6069 | .5232 $\pm$ .1551; .6063 | .3973 $\pm$ .1937; **.6070** |
| OBQA* | .3940 | .3972 $\pm$ .0048; **.4020** | .3928 $\pm$ .0044; .3980 |
| PIQA* | .7682 | .7675 $\pm$ .0082; .7731 | .7712 $\pm$ .0023; **.7737** |
| SIQA | .4744 | .4685 $\pm$ .0097; **.4770** | .4739 $\pm$ .0013; .4754 |
| WG | .6875 | .6876 $\pm$ .0025; **.6914** | .6831 $\pm$ .0032; .6859 |

## 4.2 Compression Robustness

Next, we apply GPTQ 4-bit quantization to both the pre-trained baseline and each fine-tuned model, and compare accuracy before and after compression. The maximum accuracy achieved by our pipeline exceeds the pre-trained GPTQ baseline across all eight benchmarks, with gains ranging from 0.3% to 2.4%—an improvement of the same order of magnitude as the total quantization-induced accuracy decay (approximately 1–4% depending on benchmark). Table 5 reports mean and standard deviation across five HPO replicates, together with the post-compression accuracy delta $\Delta$ relative to pre-compression mean, which measures quantization-induced degradation. Muon-optimized models exhibit smaller degradation ($\Delta$) than Adam-optimized models on six of eight benchmarks, and achieve the highest maximum post-compression accuracy on five of eight. The GPTQ-alone column serves as an ablation of the entire fine-tuning and synthetic-data component: it uses no training data and represents the cost of post-training quantization without any recovery. Our fine-tuned pipeline recovers all or most of this cost on every benchmark, with Muon consistently recovering more than Adam. We investigate the mechanism underlying this optimizer-dependent robustness in Section 4.4.

## 4.3 Hyperparameters

We observe that across tasks, our hyperparameter optimization consistently chooses to drop the Cross Entropy term entirely from the loss function for low-rank adapted models (see table 6). The consistent choice of

Table 5: Comparison of Pre-Trained-GPTQ and Fine-Tuned-GPTQ accuracies (mean ± std and max over 5 HPO runs; Δ from pre-compression mean).

| Task | Llama3.2-3B-GPTQ (Δ) | Adam-LoRA-GPTQ mean ± std max (Δ) | Muon-LoRA-GPTQ mean ± std max (Δ) |
|---|---|---|---|
| ARC-e* | .6932 (-.0168) | .6793 ± .0037 .7058 (-.0158) | .7004 ± .0059 **.7071 (-.0102)** |
| CQA | .7265 (-.0106) | .7265 ± .0043 **.7314 (-.0104)** | .7081 ± .0087 .7199 (-.0270) |
| HS* | .7064 (**-.0098**) | .6993 ± .0065 .7060 (-.0119) | .7037 ± .0062 **.7095 (-.0115)** |
| MMLU | .5787 (-.0282) | .5059 ± .1456 .5853 (-.0173) | .3818 ± .1860 **.5886 (-.0155)** |
| OBQA* | .3800 (-.0140) | .3916 ± .0057 .3960 (-.0056) | .3884 ± .0097 **.4040 (-.0044)** |
| PIQA* | .7622 (**-.0060**) | .7615 ± .0110 **.7704 (-.0060)** | .7640 ± .0028 .7682 (-.0072) |
| SIQA | .4765 (+.0021) | .4708 ± .0095 **.4811 (+.0023)** | .4752 ± .0038 .4800 (+.0013) |
| WG | .6748 (-.0127) | .6747 ± .0084 .6827 (-.0129) | .6738 ± .0124 **.6835 (-.0093)** |

Table 6: Quasi-optimal hyperparameters from a single LoRA HPO Replicate. *Note: Hyperparameters optimized via Optuna HPO using 16 samples*

| Task | Optimizer | Rank | LoRA Scale | LearningRate | Distill $\alpha$ | Distill T | W Decay | EvalLoss |
|---|---|---|---|---|---|---|---|---|
| ARC-e | Adam | 48 | 0.5 | 3.77e-4 | 1.0 | 0.51 | 0.01 | 0.079 |
| ARC-e | Muon | 16 | 1.5 | 4.17e-4 | 1.0 | 2.01 | 0.02 | 0.950 |
| CQA | Adam | 64 | 2 | 4.99e-4 | 1.0 | 0.51 | 0.04 | 0.115 |
| CQA | Muon | 24 | 1.75 | 4.91e-4 | 1.0 | 0.51 | 0.1 | 0.115 |
| HS | Adam | 24 | 1.75 | 2.34e-4 | 1.0 | 0.51 | 0.08 | 0.146 |
| HS | Muon | 48 | 2 | 4.94e-4 | 1.0 | 6.01 | 0.03 | 1.947 |
| MMLU | Adam | 24 | 1.0 | 7.56e-4 | 1.0 | 0.51 | 0.03 | 0.173 |
| MMLU | Muon | 32 | 1.0 | 5.92e-5 | 1.0 | 1.01 | 0.09 | 0.389 |
| OBQA | Adam | 48 | 1.75 | 1.35e-4 | 1.0 | 0.51 | 0 | .110 |
| OBQA | Muon | 56 | 2 | 4.96e-4 | 0.7 | 0.51 | 0.07 | 1.716 |
| PIQA | Adam | 8 | 0.5 | 1.22e-4 | 1.0 | 6.01 | 0 | 2.666 |
| PIQA | Muon | 32 | 2 | 2.61e-4 | 1.0 | 6.51 | 0.07 | 2.106 |
| SIQA | Adam | 56 | 1.75 | 1.79e-4 | 1.0 | 6.51 | 0.08 | 2.258 |
| SIQA | Muon | 24 | 2 | 3.73e-4 | 0.5 | 0.51 | 0.07 | 1.716 |
| WG | Adam | 24 | 1.5 | 3.42e-4 | 1.0 | 8.01 | 0.03 | 1.844 |
| WG | Muon | 24 | 1.75 | 3.22e-4 | 1.0 | 6.01 | 0.01 | 2.049 |

distillation $\alpha = 1$ indicates that KL Divergence between $T_2$ and $S_1$ minimizes the training loss, justifying our choice to add KL Divergence from the T2 teacher to the pipeline.

### 4.4 Outlier Analysis

We performed additional pre-quantization analysis of Adam- and Muon-optimized models to investigate the mechanisms underlying the observed difference in quantization robustness. For each task, we computed the layer-wise kurtosis of weights and activation channels using 128 calibration samples, then aggregated across layers using parameter-count-weighted averaging. Results are reported in Table 7.

Table 7: Weighted mean layer-wise kurtosis of weights (W) and activation channels (A): Adam vs Muon measured with 128 samples of calibration data

| Task | W-Kurt Adam | W-Kurt Muon | $\Delta$W | A-Kurt Adam | A-Kurt Muon | $\Delta$A |
|---|---|---|---|---|---|---|
| arceasy | 4.6038 | 4.6038 | 0.0000 | 159.3506 | 160.5997 | 1.2491 |
| cqa | 4.6038 | 4.6038 | 0.0000 | 157.4510 | 157.4692 | 0.0182 |
| hellaswag | 4.6038 | 4.6038 | 0.0000 | 157.4741 | 158.2883 | 0.8142 |
| mmlu | 4.6036 | 4.6036 | 0.0000 | 151.1230 | 148.5278 | -2.5953 |
| openbookqa | 4.6038 | 4.6038 | -0.0000 | 155.7417 | 155.1853 | -0.5565 |
| piqa | 4.6038 | 4.6038 | -0.0000 | 159.4547 | 159.3097 | -0.1449 |
| siqa | 4.6038 | 4.6038 | -0.0000 | 154.8305 | 153.7156 | -1.1149 |
| winogrande | 4.6038 | 4.6038 | -0.0000 | 161.2622 | 161.3404 | 0.0782 |

**Weight distributions are nearly identical across optimizers.** Weight kurtosis values differ by less than $10^{-4}$ in all eight tasks, which is expected: LoRA fine-tuning modifies only a low-rank subspace of each weight matrix, leaving the bulk of the pretrained weight distribution unchanged. This finding confirms that any difference in quantization robustness between Adam- and Muon-optimized models does not arise from differences in the weight value distributions themselves.

**Activation distributions show small, optimizer-dependent differences.** In contrast to weights, channel-level activation kurtosis—a measure of tail heaviness that is directly linked to quantization rounding error (Park et al., 2025)—differs non-trivially between optimizers, despite near-identical weights. On four benchmarks (MMLU, OBQA, PIQA, SIQA), Muon-optimized models exhibit lower activation kurtosis than Adam-optimized models, indicating fewer activation-channel outliers and correspondingly smaller rounding errors during 4-bit quantization. On two benchmarks (ARC-e, HellaSwag), Adam-optimized models exhibit slightly lower activation kurtosis, while CQA and WinoGrande are effectively tied ($|\Delta A| < 0.1$).

**Interpretation.** The pattern in activation kurtosis is directionally consistent with the quantization robustness results in Table 5, where Muon-optimized models degrade less under GPTQ on six of eight benchmarks. However, the correlation is not perfect: ARC-e is one of the benchmarks where Muon still shows stronger quantization robustness despite slightly higher activation kurtosis. This suggests that activation kurtosis is a contributing but not exhaustive predictor of quantization robustness in the fine-tuning regime. Other factors—including the geometry of the loss landscape, gradient trajectory during LoRA training, and how the merged low-rank update interacts with the quantization grid—likely also play a role. We view this analysis as a first step toward mechanistically understanding why optimizer choice affects post-training quantization robustness, and leave a more complete causal account to future work.

Taken together, the near-identical weight distributions and partially different activation distributions suggest that Muon's effect on quantization robustness operates primarily through the model's *inference dynamics* rather than its weight magnitudes. This is a meaningful distinction: it implies that the robustness benefit of Muon cannot be easily replicated by post-hoc weight smoothing (e.g., SmoothQuant), and instead reflects something intrinsic to the optimizer's trajectory during training.

## 4.5 Throughput and Memory

Table 8: Throughput and latency comparison: Pre- and Post- Quantization. Used to plot figure 3 *Note: Metrics measured on 1x Ampere A40 GPU, deployed with vLLM, using 1000 prompts with input 1024 length, output length 1024, max concurrency 8*

| Model | Size (GB) | Throughput (Tok/s) | TPOT (ms/tok) | ITL (ms/tok) |
|---|---|---|---|---|
| Pre-Quant | 6.01 | 1387.64 | 17.49 | 17.54 |
| **Post-Quant** | **2.86** | **1722.82** | **8.82** | **8.82** |

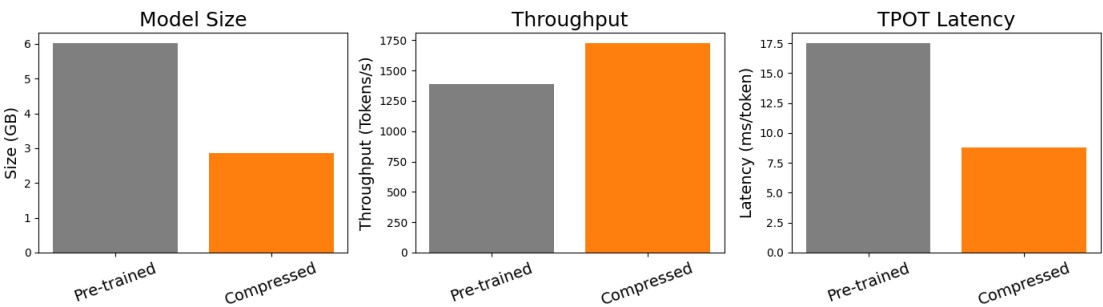

Figure 3: Throughput Comparison: Pre- and Post- Compression

We provide vLLM-deployed throughput metrics (Time per Output Token (TPOT), Intertoken Latency (ITL), and output token Throughput) for both full-precision and W4A16 quantized models in table 4.5, and plotted in figure 3. We see a decrease of 50% time for TPOT and ITL, as expected. Throughput gains are more modest than TPOT and ITL suggest, despite the faster generation, due to per-sequence prefill overhead using MarlinLinearKernel in vLLM.

## 5 Conclusion

In this work, we introduced an end-to-end framework that simultaneously specializes and compresses small LLMs for resource-constrained edge deployment. By orchestrating (1) high-quality synthetic data generation via Self-Instruct with a powerful teacher $T_1$, (2) logit-based knowledge distillation from a tokenizer-aligned teacher $T_2$, (3) parameter-efficient fine-tuning with LoRA, (4) Bayesian hyperparameter optimization, (5) the Muon optimizer, and (6) GPTQ post-training quantization, we deliver higher accuracy than naïve GPTQ quantization alone while achieving $\sim 2\times$ memory reduction and up to 50% lower per-token latency.

Beyond these empirical gains, our results support a broader view of LLM deployment as an optimization process over both task distributions and hardware constraints. Rather than treating compression and specialization as isolated steps, our framework demonstrates that jointly optimizing data generation, training objectives, optimizer dynamics, and post-training compression yields models that are more robust to downstream deployment conditions.

Our most notable findings are twofold. First, hyperparameter optimization systematically eliminates the supervised cross-entropy term ($\alpha = 1$) across tasks, revealing that pure KL-divergence alignment with a strong teacher $T_2$ is the dominant signal for minimizing loss on synthetic distillation data. Second, and more importantly, we extend recent observations about Muon's quantization robustness from pre-training to the fine-tuning regime: even a single epoch of Muon-optimized LoRA fine-tuning (combined with distillation loss) yields models that degrade less under aggressive 4-bit quantization than their Adam-optimized counterparts across 5 benchmarks. This effect is strong enough that Muon-optimized models achieve higher final accuracy than Adam-optimized ones on the majority of benchmarks despite identical architecture and compression level.

These results underscore a broader insight: optimizer choice is not merely a training detail but a critical design lever in compression-aware fine-tuning pipelines. Ultimately, our framework demonstrates that high-performing, task-specialized, edge-ready LLMs can be produced through a unified and reusable optimization pipeline, enabling rapid adaptation across diverse tasks and heterogeneous deployment environments.

**Limitations and ongoing work.** Our study is deliberately scoped to a single student architecture (Llama 3.2-3B) and a single post-training quantization method (GPTQ 4-bit) in order to obtain multi-seed statistical reporting and a controlled comparison of optimizers under matched HPO budgets. Two extensions are currently in progress and will strengthen the claims in a subsequent revision: (i) a full-rank fine-tuning

ablation comparing LoRA to unconstrained parameter updates under both optimizers, which is partially complete at the time of this submission due to compute constraints during the revision window; and (ii) evaluation across additional student architectures (e.g., Mistral, Phi). We view the present work as establishing the core empirical finding - that Muon's quantization robustness extends from pretraining into the LoRA fine-tuning regime - with architectural and full-rank generalization as the natural next steps.

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
