# OpenReview forum: "Advancing Model Refinement: Muon-Optimized Distillation and Quantization for LLM Deployment"
_TMLR — Rejected by TMLR_

### Review · Reviewer_YGRq · 2026-02-20

**Summary Of Contributions:**

The paper proposes an end-to-end framework for deploying LLMs on resource-constrained edge devices. The pipeline integrates several techniques: data distillation, knowledge distillation, LoRA, and GPTQ 4-bit quantization, Bayesian Optimization for hyperparameter tuning, and Muon optimizer which is more robust to quantization.



Pros:
* Systematic integration of multiple established compression techniques into a coherent pipeline under edge deployment constraints
* Novel empirical investigation of Muon optimizer in the context of LoRA fine-tuning + distillation + quantization
* Comprehensive evaluation across 8 standard benchmarks

Cons:
* Limited novelty: primarily engineering contribution combining existing techniques
* Modest improvements: often <1% absolute accuracy gain over GPTQ baseline
* Missing ablation studies: cannot determine which components contribute most
* Weak baselines: Did the "GPTQ Alone" baseline see the fine-tuning data? If not, The fine-tuned model should win simply because it has seen the task data. The only valid scientific comparison in the paper is "Adam-optimized LoRA" vs. "Muon-optimized LoRA."

**Audience:**

Yes

**Audience Explanation:**

- It is a timely topic, LLM compression and edge deployment are practically important and actively researched. The Integration of multiple techniques addresses real deployment challenges.
- Extension of Muon optimizer research to the fine-tuning + quantization regime is novel and interesting.

**Claims And Evidence:**

No

**Claims Explanation:**

Supported Claims:
- 2× memory compression, throughput improvements; Muon shows lower quantization degradation.

Unsupported Claims:
- The authors claim the framework demonstrates "Empirical results demonstrate the framework’s superior performance... compared to GPTQ quantization alone." But the system combines so many techniques. There is no ablation studies to isolate component contributions. Also, the results often show only <1% absolute accuracy gain over GPTQ baseline. It is unclear to me whether the "GPTQ Alone" baseline is fine-tuned on task data.

**Requested Changes:**

Critical:
- Reframe the contribution by fairly comparing with GPTQ baseline, or ablate which component contributes the most. Is it Muon-optimized distillation better than Adam-optimized distillation?

Optional but recommend:
- Compare to AWQ, SmoothQuant, QLoRA, and other recent compression methods.
- Test on Mistral, Phi, or other architectures to demonstrate generalization

---

### Review · Reviewer_AUk2 · 2026-03-08

**Summary Of Contributions:**

This is a system paper proposing a framework to train a budget model for edge devices. The proposed framework includes knowledge distillation with LoRA, quantization and Muon optimizer. Empirical evaluation demonstrates the capability of the proposed framework, where the memory is compressed to be 50 percent less than the original model. In addition, it also shows the high performance compared to post-quantization method alone.

**Additional Comments:**

- The citation style in the paper does not follow the standard citation style in TMLR. All the citations in the paper are in \citet{}, meaning mentioning authors, while most of them should be in \citep{}, meaning mentioning papers or work.
- The citation in the first paragraph of Section 2.1 is even stranger. For example: "...quantization. Sander et al. (2025) Pruning...". Is the citation "Sander et al. (2025)" supports the sentence before it or the sentence it is in. If it is the latter, should it be like: "...Pruning (Sander et al., 2025)..."
- In Section 3.1, there are quotes that are copied from LLMs without converting to the correct version in LaTEX. For example: "something" should be ``something''.

**Audience:**

No

**Audience Explanation:**

The paper is a system paper, which integrates known method into a single framework. It does not provide any new finding or insights into the problem of compressing or quantizing models in general.

**Claims And Evidence:**

No

**Claims Explanation:**

The paper claims about data distillation as part of their contribution. However, the process described in Section 3.1 (page 4) is not data distillation, but synthetic data generated from a large open-or-closed-source LLM. It only satisfies the first property to be data distillation. To be a data distillation, it should also show that the data generated is close to the original one. Here, there is no original one from T1. Hence, it is important to show that the generated data will be representative for the training data of T1.

**Requested Changes:**

The paper should make it clear about the data distillation. In its current form, it is a prompt engineering to generate synthetic data. The data generated need to be proved as being representative to achieve high performance as the teacher model T1.

Section 3.2 could be clarified further to explain what is optimized. For example, the constrained optimization presented in the section is carried out w.r.t. which variables. In addition, the loss function should be defined explicitly and may be referenced to the one in Eq. (5).

The comparison in section 4 could be improved. For example, it is unclear how the baseline GPTQ is trained for the evaluation. Is it being trained on the same data like the proposed method (task-specific data + synthetic data generated by T1)? This will ensure the evaluation is fair and meaningful.

More importantly, the paper should perform some analytical analysis to understand why the proposed framework is effective. In its current form, it is simply stacking many methods into a single framework and train a model.

---

### Review · Reviewer_xLGR · 2026-03-13

**Summary Of Contributions:**

This work aims to address the problem of deploying large language models under resource constraints. The authors propose an integrated solution by combining several orthogonal techniques: (1) synthetic data generation via a Self-Instruct-style pipeline using a strong teacher model (i.e., data distillation), (2) logit-based knowledge distillation from a tokenizer-aligned teacher, (3) LoRA-based parameter-efficient fine-tuning, (4) Bayesian hyperparameter optimization, (5) the Muon optimizer, and (6) GPTQ post-training quantization. These techniques are integrated into a unified pipeline to achieve efficient model specialization and compression. The approach is evaluated on eight standard benchmarks. The authors argue that pure KL-divergence alignment with a strong teacher is the dominant signal for minimizing loss on synthetic distillation data, and that using the Muon optimizer during fine-tuning improves robustness to subsequent quantization.

**Audience:**

No

**Audience Explanation:**

No. While the general topic of efficient LLM deployment is certainly relevant, the paper in its current form does not provide a sufficiently clear, novel, or well-validated takeaway that would be broadly useful to TMLR’s audience. The main contribution is framed as an integration of several existing techniques, but the manuscript does not convincingly isolate what is new or where the reported gains actually come from. The experimental validation is limited to a single student model scale, lacks multi-seed results and uncertainty estimates, and does not include the ablations needed to separate the effects of synthetic data generation, knowledge distillation, LoRA, hyperparameter optimization, and the Muon optimizer. In addition, the presentation is often unclear and disorganized, which makes the findings difficult to assess and further reduces their value to readers. As a result, although the problem setting is relevant, the paper does not yet offer a robust or sufficiently informative contribution for TMLR’s audience.

**Broader Impact Concerns:**

More broadly, the writing is often awkward and inconsistent, which significantly hurts clarity and at times gives the impression of heavily AI-assisted drafting.

**Claims And Evidence:**

No

**Claims Explanation:**

The paper attempts to cover too many components, but the actual presentation is incomplete and often confusing.

- In the related work section, the paper does not adequately discuss prior work on data distillation (i.e., synthetic data generation). It also does not cover related work on hyperparameter optimization. Knowledge distillation is itself a form of model compression, yet the paper presents it as a parallel category rather than situating it properly within the model compression literature.

- The experimental evaluation is too limited. The framework is evaluated only on a single student model scale, namely Llama 3.2-3B. Many implementation details are also insufficiently specified. In particular, the authors should provide multi-seed results and confidence intervals for the main accuracy comparisons, especially those in Tables 3 and 4.

- The ablation study is insufficient. Since the pipeline contains many moving parts, it is important to disentangle where the gains come from. The current experiments do not isolate the effects of synthetic data generation, knowledge distillation, LoRA, hyperparameter optimization, or Muon independently. Without such ablations, it is difficult to determine whether the reported improvements come from the optimizer choice, the task-specific synthetic data, or simply additional fine-tuning on benchmark-targeted examples.

- Most of the contents of this paper are written in a very unusual and awkward way, and I would strongly recommend rewriting them.

**Requested Changes:**

I have listed the requested changes in the weakness section. Please reference it.

---

### Decision · Action_Editor_nZQ3 · 2026-05-16

**Recommendation:** Reject

**Audience:**

Yes

**Audience Explanation:**

The topic of efficient LLM deployment is relevant, and researchers could find the findings interesting. On the other hand, the contribution does not yet provide sufficient depth or generality to be of broad interest to TMLR readers.

**Claims And Evidence:**

No

**Claims Explanation:**

The authors made meaningful revisions during the review period, including multi-seed statistical reporting (5 replicates with mean ± std), clarification that the GPTQ-alone baseline has no training data access, and a new mechanistic outlier analysis (Section 4.3). However, the central claims remain insufficiently supported. The paper still lacks the ablation studies needed to isolate which component drives the reported gains—the full-rank vs LoRA ablation is acknowledged as incomplete, and there is no controlled comparison isolating the effect of synthetic data generation from knowledge distillation. The improvements over the Adam-LoRA baseline remain modest (<1% absolute on most benchmarks), and the paper evaluates only a single architecture (Llama 3.2-3B), providing no evidence of generalization.